# High-Resolution Melting (HRM) Analysis for Rapid Molecular Identification of *Sparidae* Species in the Greek Fish Market

**DOI:** 10.3390/genes14061255

**Published:** 2023-06-12

**Authors:** Evanthia Chatzoglou, Nefeli Tsaousi, Apostolos P. Apostolidis, Athanasios Exadactylos, Raphael Sandaltzopoulos, Ioannis A. Giantsis, Georgios A. Gkafas, Emmanouil E. Malandrakis, Joanne Sarantopoulou, Maria Tokamani, George Triantaphyllidis, Helen Miliou

**Affiliations:** 1Laboratory of Applied Hydrobiology, Department of Animal Science, School of Animal Biosciences, Agricultural University of Athens, 11855 Athens, Greece; n.tsaousi@aua.gr (N.T.); emalandrak@aua.gr (E.E.M.); gvtrianta@aua.gr (G.T.); elenmi@aua.gr (H.M.); 2Laboratory of Fish & Fisheries, Department of Animal Production, School of Agriculture, Faculty of Agriculture, Forestry and Natural Environment, Aristotle University of Thessaloniki, 54124 Thessaloniki, Greece; apaposto@agro.auth.gr; 3Hydrobiology-Ichthyology Lab, Department of Ichthyology and Aquatic Environment, School of Agricultural Sciences, University of Thessaly, 38446 Volos, Greece; exadact@uth.gr (A.E.); gkafas@uth.gr (G.A.G.); saradopo@uth.gr (J.S.); 4Department of Molecular Biology and Genetics, Democritus University of Thrace, 68100 Alexandroupolis, Greece; rmsandal@mbg.duth.gr (R.S.); tokamanimaria@hotmail.com (M.T.); 5Department of Animal Science, Faculty of Agricultural Sciences, University of Western Macedonia, 53100 Florina, Greece; igiants@agro.auth.gr

**Keywords:** mtDNA, barcoding, *Pagrus*, *Dentex*, fish mislabeling, authentication, fraud, *COI*, *16s*, *cytb*

## Abstract

The red porgy (*Pagrus pagrus*) and the common dentex (*Dentex dentex*) are Sparidae species of high commercial value, traded in the Greek market. In some cases, fish species identification from Greek fisheries is difficult for the consumer due to the strong morphological similarities with their imported counterparts or closely related species such as *Pagrus major*, *Pagrus caeroleustictus*, *Dentex gibbosus* and *Pagellus erythrinus*, especially when specimens are frozen, filleted or cooked. Techniques based on DNA sequencing, such as *COI* barcoding, accurately identify species substitution incidents; however, they are time consuming and expensive. In this study, regions of mtDNA were analyzed with RFLPs, multiplex PCR and HRM in order to develop a rapid method for species identification within the Sparidae family. HRM analysis of a 113 bp region of *cytb* and/or a 156 bp region of *16s* could discriminate raw or cooked samples of *P. pagrus* and *D. dentex* from the aforementioned closely related species and *P. pagrus* specimens sampled in the Mediterranean Sea when compared to those fished in the eastern Atlantic. HRM analysis exhibited high accuracy and repeatability, revealing incidents of mislabeling. Multiple samples can be analyzed within three hours, rendering this method a useful tool in fish fraud monitoring.

## 1. Introduction

Fish is considered a valuable nutritional resource containing high content of beneficial lipids, high-quality proteins and vitamins, being a very important component in the Mediterranean diet [1]. The health benefits of fish are reflected by the increase in its global consumption with the international trade of fisheries and aquaculture products growing significantly in recent decades, expanding over continents and regions [2]. At the same time due to the complex and valuable supply chains of this commodity, there is a potential increase in fish mislabeling and/or fraud [3]. Fish mislabeling is defined as an inaccurate labeling of a specimen’s species name, weight and geographic origin [4]. Economic profit by the difference in price is the main criterion of intentional fraud by the substitution of expensive species for cheaper ones, while consumers are exposed to the risk of buying harmful products containing allergens [5] or endangered species are threatened since the substitute is exploited without reporting, thus undermining conservation efforts by supporting unsustainable or illegal fishing activities [6,7]. Fish are sold as whole or filleted (fresh or frozen) and processed (dried, salted, smoked, canned, etc.) in the markets, as well as served cooked in restaurants. Under these conditions, their morphological characteristics are altered; thus, identification of the species is often not possible, raising opportunities for fraudulent operations [8,9].

Greek consumers have a strong preference for fresh and wild fish compared to reared, frozen tinned or canned fish, whereas in many cases, the ratio price/quality is the key factor influencing consumers’ choice [10]. Two members of the Sparidae family, the red porgy *P. pagrus* and the common dentex, *D. dentex*, are widely traded species in the Greek market and are frequently served at restaurants, where fresh fish are sold under high prices, ranging from 25 to 40 EUR/kg in fish markets up to more than 60 EUR/kg in restaurants.

*P. pagrus* is a benthopelagic species, with its wild populations located in the Mediterranean Sea but also on the Atlantic coasts of America, South Europe and Africa [11,12]. Its high commercial value raises as its size increases, with European regulations setting the minimum size (total length) of its catch in the Mediterranean at 18 cm (EC), No. 1967/2006 [13]. During the previous years, due to increased consumer demand, many fish farms in Greece, Italy, Cyprus and Croatia have included the genus *Pagrus* in their production [14,15]. For many years, *Pagrus* fish farming products were sold in the Greek market under various names, and, recently, Greece has requested to add the species red seabream (*P. major*), also known as Japanese sea bream, in Annex IV of the Regulation (EC) No. 708/2007 [16], as the species has been used in Greek aquaculture for a long time. Therefore, in the Greek market regarding the genus *Pagrus,* fresh fish from (a) domestic catch, (b) Greek fish farms and (c) lagoon fish traps (the so-called “Divaria”) and frozen imported samples are available. Species of the same genus worldwide, such as the Japanese seabream (*P. major*), the blue-spotted sea bream (*P. caeruleostictus*), the red-banded sea bream (*Pagrus auriga*) and the silver sea bream (*Pagrus auratus*), share similar morphological characteristics, mainly the color and the shape. As a result, incidents of mislabeling have occurred by the mixing of the fish and their sale under the same name [17]. For example, in the German market, it has been observed that *P. pagrus*, *P. caeruleostictus* and *P. auratus* are commonly sold as “Dorade Rose” [18]. 

*D. dentex* is a demersal fish and high-trophic-level predator of the Sparidae family living in various habitats [12,19]. Its populations are distributed in the Mediterranean and eastern Atlantic. Its length reaches 100.00 cm, and it is considered the “Queen of fish” in the Greek market, where it can be found mainly as whole fresh fish catch, sliced or filleted and, in some markets, frozen. *D. dentex* is mainly fished by small-scale fisheries and leisure fishers or divers and, according to the IUCN Red List of Threatened Species 2014, is categorized as “vulnerable” [20]. There have been attempts to rear *D. dentex*, some of them with successful outcomes [21,22], as well as the species *D. gibbosus*, which shares strong similarities with both *D. dentex* and *P. pagrus* and is considered one of the most delicious marine fish. *D. gibbosus* resembles and is often marketed as *D. dentex* or *P. pagrus*, with taste and appearance that are hardly distinguishable by consumers [23]. The higher commercial value of wild *P. pagrus* and *D. dentex* from wild catches compared to imported, reared counterparts or other species of the Sparidae family (i.e., *D. gibbosus*, *P. erythrinus*) raises the need for accurate identification and labeling of the fish according to EU 1379/2013 guidelines [24]. 

Traditionally, fish species identification is based on external morphological features, including shape, various relative measurements of body parts and otoliths [25]. For many years, protein-based methods have been used in official laboratories, such as protein isoelectric focusing (IEF) of soluble muscle proteins [18] or the enzyme-linked immunosorbent assay (ELISA) [26]; however, these techniques are restricted to the identification of fresh samples, as they cannot be applicable when fish are filleted, canned or cooked due to the removal of morphological characteristics and denaturing of proteins [25]. On the other hand, DNA-based methods have proven to give accurate and reliable results and are more rapid and cost effective than protein-based analyses [8,9,25,26]. The principle of these methods is based on species-specific polymerase chain reaction (PCR) coupled with DNA sequencing [27,28] or RFLPs [17], SSCP [18,29] and/or real-time (qPCR) coupled with end-point analysis and high-resolution melting (HRM) [26,30,31,32,33]. Mitochondrial DNA is often used in fish identification as it has features that facilitate the analyses: (a) lack of introns, (b) a large number of copies, (c) maternally inherited, (d) no recombination and (e) circular structure [29]. Moreover, it evolves much faster than nuclear DNA and thus enables even closely related species to be differentiated and identified [34].

DNA barcoding is a universal taxonomic method that refers to the sequence of a 650 bp fragment of the *COI* gene amplified with universal primers and is used to identify it as belonging to a particular species [8,27,35]. A large number of standardized reference DNA sequences is collected in databases such as GenBank [36] and the Fish Barcode of Life (FISH-BOL) campaign [27]. By using these libraries, an unknown fish sample can be matched against fish species reference sequences to determine its species [27]. This approach has gained much attention in recent years since the survey of authenticity and single laboratory validated method for DNA barcoding for the species identification of fish has been adopted by the FDA for the identification of fish products [37]. Other mtDNA genes that are used in PCR-sequencing-based methods as species-specific markers are *cytb* and *16s*, under the term forensically informative nucleotide sequencing (FINS) [29]. These genes exhibit high levels of variation between species but low variations within species [9,38].

DNA-sequencing-based methods are reliable, with the capability to accurately detect cases of substitutions [35,39]; however, they are expensive and time consuming and thus not easily applicable to extensive market controls [30,32,33]. Conventional PCR-based methods are used for fish species identification, such as PCR-RFLP [40,41] and multiplex PCR [25,29], which rely on the analysis of species-specific band patterns by agarose gel electrophoresis. Real-time PCR technology employs fluorescent dyes that allow for the direct observation of results in real-time and has paved the way for the creation of post-amplification analysis techniques such as HRM based on the quantitative analysis of the melting temperature (Tm) curve of an amplified DNA fragment [26,42,43]. HRM has been used for fish species identification of fresh [32,44] or processed samples [30,33,45].

In this study, DNA-sequencing-based methods (COI barcoding), conventional PCR methods (PCR-RFLP and multiplex PCR) and real-time PCR coupled with HRM analysis are used on raw, frozen and cooked samples for the identification of *P. pagrus* and *D. dentex* from Greek fisheries, and they are distinguished from Pagrus aquaculture specimens, their imported counterparts or other Sparidae species that are sold in the Greek market and could be used in mislabeling incidents. The aim of this study is to develop an accurate, rapid and cost-effective method that could be used in extensive market controls to detect mislabeling and prevent fraud incidents.

## 2. Materials and Methods

### 2.1. Samples Collection and Morphological Identification

The 106 *Pagrus* spp. samples used in this study were collected from (a) Greek fisheries, (b) Greek fish farms, (c) lagoon fish traps in Greece, (d) imported fish sold at the Greek market and (e) restaurants. The 57 *D. dentex* specimens were collected from Greek fisheries (sampled), and 1 was imported from the Mediterranean Sea (Tunisia). Moreover, three identified samples of *P. pagrus* and three of *D. dentex* from the east coasts of Spain were donated to be used as controls in our experiments. Other Sparidae species incorporated in this study were *D. gibbosus, Dentex angolensis* and *P. erythrinus* from Greek fisheries and/or imported and sold in the Greek market. Most of the imported species were labeled with origin from the Atlantic Ocean fished in Senegal or Argentina. The areas of collection are shown in Figure 1, and the number of samples tested from each region collected is shown in Table 1.

When possible, the specimens were identified by their external morphological characteristics (body shape, fins, rays and color) according to FAO [46] and categorized into groups by their origin. The difference between *P. pagrus* and *P. major* is the absence of a white line at the end of the caudal fin and the shorter pectoral fin in the latter [12]. Between young *Pagrus* spp. and *P. erythrinus,* the differences are found in body shape, color and lateral fins (Figure 2). *P. caeruleostictus* shares many common characteristics with *P. pagrus* and *P. major*, being distinguished by the presence of blue spots on the upper body, a dark spot at the end of the dorsal fin and its elongated 3rd ray [47]. *D. dentex* has a compact oval body [48], with characteristic canine teeth and dorsal spots that are present in bigger fish. Regarding species of the same family but of a different genus, the main characteristic of *D. gibbosus* is that the first two rays of the dorsal fin are very small but the third and fourth are distinctively enlarged, a feature that sets it apart from other species [12,48]. *P. erythrinus*’s body shape is oval and laterally flattened. Its body color is red without stripes, the snout is at least twice as long as the eye diameter (Figure 2), and its caudal fin is forked with a red spot on its base [12,48].

For ten samples, morphological characterization was not possible as they were either tissue samples from recreational fishers, collected as filleted *P. pagrus* from the Greek fish market or cooked and served at Greek restaurants (Figure 2b). Moreover, five imported samples were frozen; therefore, some of the morphological characteristics were altered. Samples from all species and origins already identified were cooked in various ways, roasted, fried or boiled in soup with other ingredients, in order to estimate the efficacy of different methods when compared with the respective raw samples (Table 1). After morphological identification, tissues were collected from all samples, and whole fish samples and isolated tissues in 70% ethanol were stored at −20 °C.

### 2.2. DNA Extraction and PCR 

For each fresh sample, 25 mg of tissue (muscle or liver) was dissected with sterile disposable plastic forceps, diluted in PBS and homogenized with a cordless motor pellet pestle (Kimble). Total DNA was extracted using the NucleoSpin^®^ Tissue (Macherey-Nagel) kit for all fresh tissues and the NucleoSpin^®^ Food (Macherey-Nagel) kit for cooked or frozen samples and small amounts of tissue, according to the manufacturer’s instructions. DNA concentration and quality were assessed using Biophotometer D30 (Eppendorf) and run on 0.8% agarose gel. DNA of high quality (260/280 ratio 1.8–2.0 and 260/230 ratio 2.0–2.2) from both extraction procedures was used in downstream applications, and the results were compared. Primers sourced from the literature or designed in this study with Geneious Prime [49], based on the complete mtDNA of *P. pagrus* (Accession number NC_072936) and *D. dentex* (Accession number MG727892), were used for the amplification of different mtDNA regions of the Sparidae (Table 2). All PCR reactions were performed in MiniAmp Plus Thermal Cycler (Applied Biosystems), using the KAPA Taq PCR Kit (KAPABIOSYSTEMS). In each reaction, 200 ng of total DNA was used as a template. PCR reactions were performed as follows: initial denaturation at 95 °C for 3 min, followed by 40 cycles of amplification. The annealing temperatures were adjusted according to the Tm of primers (Table 2) used in each reaction, with an elongation of 30–90 s and final elongation of 3 min. Specifically, for barcoding, a fragment of 706 bp of *COI* was amplified using the pair COIpp-COIUnR; for *cytb,* a fragment of 583 bp was amplified using the pair cytbF1-cytbUR; and for *16s,* a fragment of 681 bp was amplified using the pair 16sF2-16sF2. For these three reactions, the annealing temperature was 54 °C. Amplification conditions targeting different fragments of Sparidae mt DNA used in RFLPs, multiplex PCR and HRM analyses are described in Section 2.4, Section 2.5 and Section 2.6. For the verification of amplification efficiency, the products were run on 1.2% agarose gels stained with Midori Green (Nippon Genetics). 

### 2.3. Sanger Sequencing

For the COI barcode, all samples were sequenced for both strands using the Sanger dideoxy method by EUROFINS Genomics sequencing services. The obtained COI sequences for each haplotype present in different fishing areas or in the fish market are available in GenBank with accession numbers for *P. pagrus*: OQ208746-OQ208756, OQ211305-OQ211329, OQ272113-OQ272123, OQ865594-OQ865603, OQ208828-OQ208830, OQ281602, OQ272124, OQ281603-OQ281609, and OQ860775-OQ860777; *P. major*: OQ860773, OQ860774, OQ888163, and OQ888164; *P. caeruleostictus*: OQ861107; *P. erythrinus*: OQ861152-OQ861162 and OQ890738; *D. dentex*: OQ862795-OQ862820; *D. gibbosus*: OQ880490-OQ880493 and OQ880557-OQ880559; and *D. angolensis*: OQ888790. For cytb and 16s fragments, 165 samples representing all species listed in Table 1 were sequenced. Sequences showing different SNPs from these fragments were deposited to Genbank and are available with accession numbers for *P. pagrus* 16s OQ892284-OQ892292 and cytb OQ915022-OQ915035, as well as for *D. dentex* 16s (OQ903885-OQ903890).

### 2.4. In Silico Analysis and Detection of Polymorphisms

All sequences obtained were identified with the Blast search tool [52], compared to sequences from the GenBank [36] database or matched with reference sequences by using the BOLD database [27]. Samples were aligned for each gene region with ClustalW2 [53]. For the alignments, samples from this study, as well as sequences extracted from the databases, were used. After in silico analysis of *COI*, *cytb* and *16s* genes and total mtDNA of Sparidae species and polymorphic sites were detected. New primers were designed with Geneious Prime to amplify smaller fragments (100–150 bp) that were used in HRM experiments or fragments of different sizes to be used in multiplex PCR. For RFLP analysis, the aligned partial sequences, as well as the complete genomes of the available Sparidae species in GenBank with the respective accession numbers *P. pagrus* (NC_072936), *P. major* (NC_003196), *P. caeruleostictus* (MN319701), *P. erythrinus* (NC_037732.1), *D. dentex* (MG727892), *D. gibbosus* (NC_037731) and *D. angolensis* (NC_044097.1), were checked for restriction enzyme digestion sites with Geneious Prime.

### 2.5. Restriction Fragment Length Polymorphism (RFLP) 

For RFLP analysis, the *COI* barcode fragment (Table 1) was digested with HindIII (Takara) to distinguish between *D. dentex* and other species (Table 3) and with Sau3AI (EnzyQuest) to identify *P. pagrus*. Additionally, for *P. pagrus* and related species, the *cytb* fragment (Table 1) was digested with Sau3AI, and a 1292 bp fragment of the CR (Table 3) amplified with cytbF2 and CRR primers was digested with XbaI (Takara). 

The products were run on 1.2% agarose gels stained with Midori Green (Nippon Genetics), and the band sizes were estimated with FastGene^®^ 100 bp DNA Ladder (Nippon Genetics) and 50 bp DNA Ladder (Jena Bioscience GmbH). 

### 2.6. Multiplex PCR 

In a single reaction, a combination of primers was used to amplify fragments of different lengths. Multiplex PCR was performed using the KAPA2G Fast Multiplex PCR Kit (KAPABIOSYSTEMS) using the supplied buffer at a final concertation of 3 mM MgCl_2,_ 100 ng of the template and 0.2 μM of each primer in combinations as shown in Table 4. After amplification, the products were run on 1.5% agarose gels stained with Midori Green (Nippon Genetics).

### 2.7. HRM Analysis 

In order to define a suitable region to be used as a species-specific marker for HRM analysis, fragments of *COI* barcode, *cytb* and *16s,* were selected, as they are often used in phylogenetic studies and several reference sequences are available in GenBank. Sequences of these samples and databases were aligned with the aim to locate SNPs that could discriminate between species and/or populations within species. By these comparisons, regions that could serve as templates for smaller fragments’ amplification (113–156 bp) were defined. For the amplification of the selected regions, specific primers were designed (Table 5) that are able to anneal with all species of the Sparidae family. Different DNA template concentrations (from 1–10 ng) were tested for both isolation procedures in order to check the accuracy of the method, and the optimum concentration (10 ng) was defined. PCR conditions were adjusted to ensure amplification for all samples, and, prior to HRM analysis, the efficacy of primers was tested by conventional PCR products and run on 2% agarose gel to verify product amplification. The curve analysis parameters [43] for each fragment tested were optimized by using samples of known sequences as controls, and final values were determined when the percent confidence was 95–99.9% for all species clustering. All samples used were sequenced before or after HRM analysis for the verification of the results. 

HRM reactions were performed with KAPA HRM FAST qPCR KitAll PCR (KAPABIOSYSTEMS) in CFX96 Real-Time PCR thermocycler (BIO-RAD LABORATORIES INC). The 2× Master Mix supplied by the kit contains EvaGreen and was mixed with MgCl_2_ and the pair of primers for each gene as shown in Table 3. Each 20 μL reaction contained 5 ng of total DNA and final concentrations of 2.5 mM MgCl_2_, 200 nM for each primer. A three-step protocol was used for amplification as follows: enzyme activation at 95 °C for 2 min, followed by 45 cycles of denaturation at 95 °C for 5 s, annealing according to the set of primers (Table 3) for 20 s and elongation at 72 °C for 10 s. Melt curve dissociation was performed in 0.2 °C increments from 65 °C to 95 °C. PCR amplification and melting processes were monitored in real time by plate read through CFX manager 3.1 software (Bio-Rad Laboratories Inc. Hercules, CA, USA). Curve data were analyzed using Bio-Rad Precision Melt Analysis Software, version 1.2 (Bio-Rad Laboratories Inc. Hercules, CA, USA) [43]. Analysis options, melt curve shape sensitivity (which determines the level of stringency applied in classifying the melt curves into distinct clusters) and the Tm difference threshold (which determines the minimum melting temperature difference between samples below which the software will classify them as belonging to different clusters) [43] were adjusted for each gene to obtain maximum confidence percent values, i.e., the relative probability of the sample of being in a cluster, and were compared to sequencing results. Samples were run as triplicates in 2 or 3 independent trials.

## 3. Results

### 3.1. Morphological Identification and DNA Isolation

All whole-body samples collected from Greek fisheries (FC) and supermarkets (FM/SM) were identified by their morphological characteristics as described in Section 2.1. When small fish from Greek catch were closely examined, *P. pagrus* with a total length less than 18 cm up to 500 gr were mixed with *P. erythrinus.* In one case, morphological identification failed to identify a young *D. gibbosus* that was also mixed with *P. erythrinus.* In two cases of big fish (>4 kg) collected from the Greek market and sold as *P. pagrus*, morphological examination identified them as *D. gibbosus*, as they had the characteristic dark spot on the end of their dorsal fin, although the characteristic enlarged third and fourth rays of the dorsal fin were not visible. Therefore, it is possible that these rays were cut in order to sell the fish as *P. pagrus.* The majority of imported *P. pagrus* samples were collected from the fish market and the supermarkets as frozen and packed (Figure 2b). When fish were thawed, some characteristics were detectable; however, morphological identification was not always clear, as they were partly destroyed from freezing. When imported *P. pagrus* (PpI) are sold fresh, they are morphologically identical to *P. pagrus* from Greek fisheries. All Pp samples from Greek fish farms (AQ) as well as all samples from lagoon fish traps (LFT) labeled as *Pagrus* spp. Were identified as *P. major*. 

For *D. dentex* specimens, morphological identification was clear for smaller (1 kg) or bigger (up to 3 kg) fish. In one case, an imported frozen and packed sample was labeled both with the common name pandora but with the scientific name *D. dentex*, and when unpacked and defrosted, morphological identification showed that it was *P. erythrinus.*

### 3.2. Sequencing-Based Identification—COI Barcoding 

#### 3.2.1. *P. pagrus*

*COI* barcoding accurately identified all samples examined. For PpG (Table 1), a predominant haplotype and eleven more haplotypes with one or two SNPs were determined. Each of these barcodes shows 99–100% homology with barcodes of *P. pagrus* mined from GenBank and Bold databases corresponding to samples from different areas of the Mediterranean Sea [54,55,56,57,58]. Specimens of PpI sold in the Greek market were identified as *P. pagrus,* showing 98–100% homology with barcodes of *P. pagrus* from the eastern Atlantic Ocean, with the exception of one frozen sample labeled as *Sparus pagrus*, which was identified as *P. caeruleostictus* (Pc). In four cases of samples sold as filleted PpG, DNA barcoding identified two of them as *P. pagrus* and two as *D. gibbosus*. Two samples offered as *P. pagrus* by recreation fishermen were identified as *P. major*. Three out of seven samples collected at restaurants (RE) were offered as *P. pagrus*, but one was identified as *D. gibbosus* and two as *P. major*. Barcoding was in accordance with the founding of morphological classification in the case of two samples of big fish (>4 kg) sold as *P. pagrus* but identified as *D. gibbosus*. In addition, in ten samples of juvenile fish, the classification of which based on their morphological characteristics was not clear, barcoding confirmed that two of them were *P. erythrinus* and one was *D. gibbosus*. All samples from different Greek aquaculture farms (AQ) and lagoon fish traps (LFT) were identified as *P. major*. Barcodes of known samples that were cooked for the purposes of this study were identical to the respective raw specimens.

#### 3.2.2. *D. dentex*

Two barcodes with one SNP were found for the populations of DdG (Table 1), distributed randomly in all sampling areas. Both barcodes were compared against sequences from GenBank and Bold databases and found to be identical with barcodes from *D. dentex* samples from the Mediterranean Sea [54,59,60]. The only imported samples in the Greek market come from Tunisia (DdM), and their sequences are identical to the samples from Greek fisheries, as well as from Spain and Italy [59,60]. One incident of mislabeling was detected regarding a sample given by a recreational fisherman as *D. dentex*, which was identified as *D. gibbosus.*

#### 3.2.3. Other Sparidae Species

Twenty-five samples of *D. gibbosus*, *D. angolensis* and *P. erythrinus*, from Greek fisheries or imported and sold in the Greek market (Table 1), were identified by their morphological characteristics and by barcoding in order to be used in downstream applications as characterized species-specific controls. 

### 3.3. Non-Sequencing-Based Techniques

#### 3.3.1. RFLP Analysis

When the *cytb* fragment was digested with Sau3AI (Figure 3a), fragments of 420 bp, 90 bp and 70 bp are generated for PpG, as well as three fragments of 450 bp, 80 bp and 50 bp for PpI, two fragments of 380 bp and 200 bp for Pe, two fragments of 430 bp and 150 bp for Pc and two fragments of 420 and 160 bp for Pm, whereas no digestion sites were found for Dg (Figure 3a). When the COI barcode fragment was digested with HindIII, two bands of approx. 500 bp and 200 bp for PpG and Dg and two bands of 620 bp and 80 bp for Pm were obtained, but no restriction site for PpI was detected (Figure 3b). 

When cutting the CR fragment with XbaI, one restriction site was found, resulting in two fragments of appr 500 bp and 700 bp for PpG, whereas this site is absent from PpI, Pm, Pc and Dg. Cooked samples were also digested, giving the pattern expected for *P. pagrus* in each case. However, in some cases of frozen samples or filleted fish, the amplification of PCR products was not as efficient as expected; therefore, the products could not be easily detectable on agarose gels (Figure 3). 

The *COI* barcode fragment was also used for the discrimination of DdG compared to *D. dentex* samples fished in the Mediterranean Sea and found in the Greek market, as well as other Sparidae species. Digestion with HindIII showed a two-band pattern of 450 and 250 bp for all *D. dentex* as well as for *D. angolensis* samples but not for *D. gibbosus* and *P. erythrinus* (Figure 4). Therefore, the identification of *D. dentex* cannot be ascertained in the case *D. angolensis* samples are introduced.

#### 3.3.2. Multiplex PCR

Multiplex PCR showed that the expected three-band pattern of 461, 295 and 211 bp was observed only in *P. pagrus* but not in the other species; however, some bands were not detectable in some PpG and PpI samples (cooked or frozen) (Figure 5a), or, in some PpG samples, the presence of other bands indicates nonspecific amplification. When *D. dentex* and other Sparidae species were analyzed as described in Section 2.5, different patterns were obtained for different species and populations. For the *D. dentex* samples analyzed, only the 249 bp band was amplified in all samples. Da showed two bands at 583 and 295 bp; DgG showed two bands at 583 and 249; DgI showed all four bands; *P. erythrinus* from Greek fisheries and *P. caeruleostictus* showed three bands of 583 bp, 295 bp and 249 bp; and imported *P. pagrus* and *P. erythrinus* showed one band each of 295 bp and 249 bp. 

#### 3.3.3. HRM Analysis for the Detection of *P. pagrus* from Greek Fisheries

For the detection of the substitution of *P. pagrus* from Greek fisheries when compared to the samples listed in Table 1, three fragments (*COI*, *cytb* and *16s*) were tested by HRM, but only *cytb* (113 bp) and *16s* (156 bp) were informative for the comparison. These sequences were selected for being 100% identical in all specimens of PpG but showing SNPs with specimens of closely related species and populations of different geographic origins, as shown in Figure 6a and Figure 7a alignments. In HRM analysis, PpG and PpM (red) were assembled in one cluster that was used as the reference, distinct from the imported *P. pagrus* cluster (PpI-orange) and distinct from other species including *P. major* (Pm-blue), *P. caeruleostictus* (Pc-dark blue), *P.* (Pe-green) and *D. gibbosus* (Dg-purple) for both *cytb* and *16s* fragments, as shown in Figure 6b and Figure 7b. Cooked samples, as expected, were clustered with the same curves as the respective raw samples (Ck-shown in yellow in Figure 6c and Figure 7c). Confidence levels for all samples are given in Appendix A.

For *cytb*, the melt region was automatically set by the software, whereas, in order to achieve maximum confidence levels, the melting curve shape sensitivity and Tm difference threshold were set manually at 75% and 0.2, respectively. Under these conditions, confidence levels ranged from 97 to 98.5% for the PpG cluster. The correct clustering of 99% of tested samples was confirmed by sequencing. One unknown sample of PpG (SP) was clustered separately both from PpG/M and PpI, and sequencing results revealed one SNP between this sample and other PpG/M samples.

The *16s* melt region was automatically set by the software, whereas, in order to achieve maximum confidence levels, the melting curve shape sensitivity and Tm difference threshold were set manually at 75% and 0.2, respectively. Under these parameters, confidence levels ranged from 97.2 to 99.9% for the PpG cluster, and all samples were accurately clustered as distinct species or different geographical origin groups (i.e., PpG/M vs. PpI).

Both sets of primers used in HRM experiments amplified all tested samples; however, SNPs within the primer binding regions were higher for *cytb* primers (Figure 6a) compared to *16s* primers (Figure 7a). Moreover, the percent identity between tested groups is higher for the 16s fragment (93.9–98.7%) than cytb (81.0–96.5%) (Appendix A). Our results showed that these variations do not interfere with the correct clustering of samples for both mtDNA regions tested.

#### 3.3.4. HRM Analysis for the Detection of *D. dentex* from Greek Fisheries

For *D. dentex*, two fragments, *COI* (Figure 8) and *16s* (Figure 9), were chosen for the discrimination of DdG from other Sparidae species. The amplification of cytb fragment was not efficient for *D. dentex* specimens. The only imported samples of *D. dentex* that were available for the analysis were one sold in the Greek market from Tunisia and one sample from Spain. In HRM of *COI*, analysis parameters were adjusted manually: a pre-melt range of 73.3–73.8, post-melt range of 77.5–78.0, melting curve shape sensitivity at 75% and Tm difference threshold at 0.5. Under these parameters, confidence levels for the *D. demtex* cluster ranged from 98 to 99.6%. The 16s melt region was automatically set by the software, whereas, in order to achieve maximum confidence levels (96.0–99.8%), the melting curve shape sensitivity and Tm difference threshold were set manually at 75% and 0.2, respectively. For both *COI* and *16s* fragments, *D. dentex* samples (DdG/M-pink) were assembled in one cluster and separated from other species, as shown in Figure 8b and Figure 9b, even when cooked samples were incorporated into the analysis (Figure 8c and Figure 9c). The correct clustering of samples was in accordance with sequencing results for all tested samples. 

Primers used for the amplification of *COI* and *16s* showed mismatches in the binding region for some Sparidae species; however, all tested samples were efficiently amplified for both fragments. The percent identity of *D. dentex* compared to other species is 89.6–91.3% for the COI fragment and 90.3–94.2 % for 16s. In *16s, D. dentex* and *P. caeruleostictus* sequences have one nucleotide insertion compared to other Sparidae species. The observed sequence variations did not affect the correct clustering of the species for both mtDNA regions tested. The percentage of sequence identity for the fragments is presented in Appendix A.

## 4. Discussion

Both species, *P. pagrus* and *D. dentex*, selected for this study are widely traded and consumed in Greek fish markets and restaurants, especially during summertime when local consumers and tourists are keen on tasting high-quality fresh domestic fish. As a result of high demand, the price of fresh Greek products is increased, and substitution incidents can occur especially when fish are offered filleted or cooked and covered with sauces in a delicious dish of high price. The close morphological resemblance of *P. pagrus* and *D. dentex* with other local or imported Sparidae species of a lower price, as well as aquaculture products, all traded in the Greek markets, leads to the hypothesis that they can be used in substitution events with economic profit. 

Currently, DNA sequencing-based methods, and, more particularly, DNA barcoding, are indispensable tools for species identification, providing the highest accuracy and certainty of the results [39]. Therefore, in this study, DNA barcoding was used to precisely identify the species of all specimens that were used as reference samples in the subsequent analyses. 

A large number of barcodes are available for the Sparidae in BOLD and GenBank databases; however, there are few for *P. pagrus* and *D. dentex* from Greek seas. For *P. pagrus*, all DNA barcodes sequenced from fish caught in Greek seas show up to 100% identity with barcodes of fished individuals from Turkey [54], Israel, the Central Mediterranean, Egypt and Spain [54,55,56,57] but lower with those from the Atlantic Ocean. [58]. The same results were also obtained for *D. dentex* with 99.67–100% identity with other Mediterranean samples mined from GenBank and Bold databases [57,59,60]. The homogeneity of *P. Pagrus* populations in the Mediterranean and their differences with the Atlantic populations have been described in population studies [58,61]. The Strait of Gibraltar is a natural border between the Mediterranean Sea and the Atlantic Ocean, and several studies have shown a barrier to gene flow for several species’ populations, including *Pagrus* [61]. In this study, two samples collected from the Greek market, or served filleted at a restaurant and identified as *D. gibbosus,* were labeled and sold as *P. pagrus*. Although *D. gibbosus* is a tasty fish, appreciated by recreational fishermen, it is not as popular as *P. pagrus* or *D. dentex* to regular consumers, its price is lower and imported fish of this species are traded in the Greek market, indicating that these findings are mislabeling incidents. All samples sold as Greek aquaculture products or from lagoon fish traps were identified as *P. major*. Since *P. major* is the only *Pagrus* species in Greek aquaculture, it is clear that fish from lagoon fish traps are not wild *P. pagrus* as claimed but fish introduced from aquaculture farms. The presence of four samples of *P. major* in the Ionian Sea, near the entrance of Amvrakikos Gulf [15], a location which is in proximity to aquaculture farms and lagoon fish traps, indicates the dispersal of this non-native species in the area. In our study, two samples fished in Cyclades (CY) and Crete (CR) were offered by recreational fishermen as *P. pagrus*, but they were identified as *P. major*. The presence of *P. major* has already been reported in the Adriatic Sea [62] and eastern Mediterranean [63], but its presence is now confirmed also in the east coasts of the Greek mainland and South Aegean. The use of *P. major* as a common species in Greek porgy aquaculture, and the occurrence of this species in the Greek seas, raises the question about the environmental impact of the rearing of a non-native species. Moreover, in four cases examined in this study, these fish were sold in the market and in restaurants under the genus name without clearly identifying the species. *P. major* aquaculture products as well as LFT fish have lower prices (16–17 EUR/kg for AQ fish and 20–22 EUR/kg for LFT fish) compared to wild catches of *P. pagrus* (EUR 25–30), indicating cases of mislabeling. However, it remains uncertain whether it was unintentional, since this species is also found in the wild, or whether these were cases of fraud. For *D. dentex,* only one case of mislabeling was detected regarding an imported fish that was identified as *P. erythrinus,* and one sample donated by a recreational fisherman was erroneously identified as *D. dentex* but assigned by barcoding as *D. gibbosus*, outlining the difficulty of Sparidae species identification by morphological characteristics.

The high occurrence of mislabeling, especially for *P. pagrus,* raises the need for extensive market control. In this study, three methods (PCR-RFLPs, multiplex PCR and HRM) were used to discriminate between closely related species and between populations of different geographic origin within species, aiming to analyze a large number of samples with low cost and in a time-effective experimental procedure. Fresh, frozen and cooked samples were used to estimate the sensitivity, tolerance, efficiency and consistency of each method and finally to propose the most suitable protocol applicable to market controls. When conventional PCR methods (PCR-RFLPs and multiplex PCR) were applied, *P. pagrus* and *D. dentex* were identified compared to other Sparidae species, and the methods proved to be cost effective compared to sequencing methods. However, both methodologies require the amplification of fragments ranging from 150 to 1292 bp, and a robust PCR product is necessary in order to be visible on agarose gels. RFLPs analysis is time consuming as several steps are required—amplification, digestion reactions and two electrophoresis procedures—thus restricting the number of samples analyzed per day. Multiplex PCR is faster and cheaper than RFLPs, but nonspecific amplification was detected, and optimal PCR products were not consistently obtained for bigger fragments (>400 bp), especially when frozen or cooked samples were analyzed. This could be attributed to DNA partial degradation in fish tissues, a procedure that starts at the moment the fish die [64] and continues until the sampling time. In the case of samples collected from the market, the exact time between organism death and tissue storage, as well as conservation procedures on a vessel, is usually unknown. Although storage in ice or refrigeration delays DNA degradation, post-mortem interval can affect DNA integrity [64,65]. In the case of frozen samples, although low temperatures can slow down this procedure, long periods of storage or defrosting can affect DNA quality [64]. For cooked samples, we have to consider that when DNA undergoes thermal treatment, it can be partially degraded into fragments ranging from less than 100 bp up to about 500 bp [29] and that a number of compounds present in processed or cooked foods may act as inhibitors in PCR [34].

Since our study involves commercially available and cooked samples, in order to overcome the limitation of DNA integrity, real-time PCR coupled with HRM analysis was used. Targeting shorter DNA regions (113–156 bp) ensures amplification even if DNA is degraded, the use of next-generation fluorescence dyes such as EvaGreen gives the method high sensitivity [43], and melt curve analysis gives the possibility of discriminating between two PCR products with one SNP. 

A suitable marker for species authentication must be variable even between the closest species and display either low or ideally no intra-specific variations across the geographic distribution area. Moreover, it should be widely studied to enable comparison between the nucleotide sequence from an unknown sample against reference sequences in a database [34]. Based on the intra-species and inter-species polymorphisms of the Sparidae representatives, the selected *16s* fragment best satisfies these criteria, and the analysis method was optimized for the identification of an unknown sample either as *P. pagrus* or *D. dentex* from domestic catches. All mislabeled samples precisely identified by sequencing were equally discriminated in HRM analysis, and the best results were obtained with the 16s fragment. Amplicons were obtained consistently and effectively either from small amounts of tissues or from partially degraded samples when larger fragments failed to amplify. The clustering of each species showing characteristic peaks for each reference species was consistent regardless of the tissue used, its state or the DNA extraction protocol. The latter analysis, in the case of *D. dentex,* could discriminate the species from other representatives of the Sparidae family, whereas in the case of *P. pagrus,* it could discriminate domestic samples from other Sparidae species and from imported *P. pagrus* fished in the Atlantic Ocean. The protocols in this study are optimized for *P. Pagrus* or *D. dentex* from the Mediterranean; therefore, when an unknown sample is not clustered as such, the species can be determined by subsequent sequencing. However, our results show that by applying the same analysis, samples of *P. major* can also be identified, as they form a consistently separate and identifiable cluster. This suggests that the approach could be applied for monitoring the presence of *P. major* escapes in the Mediterranean Sea. The analysis was equally efficient even when frozen or cooked samples were tested. Therefore, boiling, frying, adding sauces or long periods of ice frosting and defrosting does not affect the amplification of the mtDNA regions tested. As in the case of DNA barcoding, HRM failed to discriminate between populations from the Greek seas and other Mediterranean Sea areas; however, fresh samples from other Mediterranean countries are not often found in the Greek market or are sold at a high price (20–25 EUR/kg) compared to the imported frozen *P. pagrus* (5–8 EUR/kg). Therefore, the possibility of using them for fraudulent actions with economic profit is low. 

HRM analysis has previously been used for fish species identification in Gadidae [33,45], hake [66], Takifugu pufferfish [31], Macrourus [32], sharks [67], pangasius [68] and sepias [69]. It has been validated for mussels [70] and salmonids [71], for which it has been applied to market surveys, showing the potential of a common approach for seafood authentication throughout the supply chain. HRM is carried out in a single reaction in one apparatus, and a large number of samples (95 max) can be analyzed within three hours, as post-PCR melt curve profiles are rapidly processed by the analysis software. The advantages of being applied on samples when no morphological identification is possible (i.e., filleted or juvenile fish) or for those that are cooked and canned [70] rapidly in real-time observation of the results in a cost-effective manner, eliminating the necessity of large-number sequencing reactions, are obvious. Moreover, using a single piece of equipment for analysis offers the advantage of enabling the technique to be established in multiple control laboratories [71] across the supply chain.

## 5. Conclusions

Fisheries play an important socioeconomic role in Greece, providing the market with wild fish of high nutritional value. In order to enable consumers to make informed choices, it is necessary to provide clear and comprehensive information about the species and validate the origin of the traded fish. European food legislation is particularly strict, and traceability systems based on product labeling are mandatory in all European countries, as described in Regulation (EU) No. 1379/2013 [24]. Labels must provide clear and accurate information on the product’s origin, species, catch or farming method and production method. In this study, incidents of intentional or unintentional mislabeling for *P. pagrus* and *D. dentex* have been identified in the Greek market, either in raw or cooked samples. In order to protect the consumers’ preference for local products, extensive sampling in fish markets and restaurants needs to be operated. We propose HRM analysis as a fast, effective, sensitive, repeatable and cost-effective method that can be applied in systematic market controls as a rapid tool against fish fraud and mislabeling detection.

## Figures and Tables

**Figure 1 genes-14-01255-f001:**
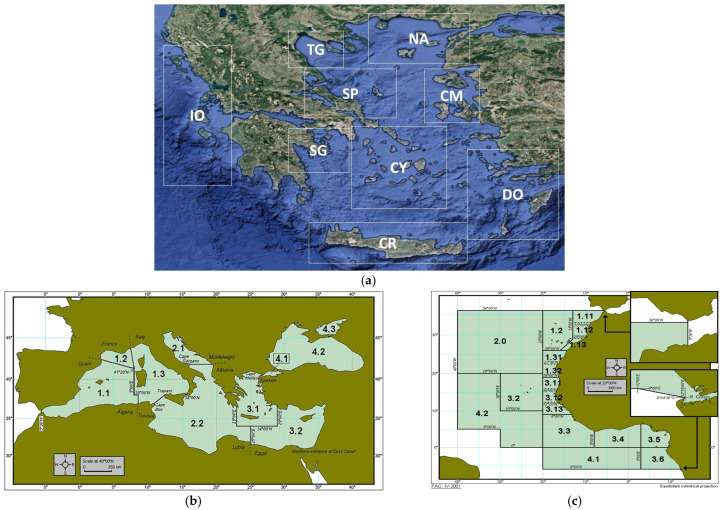
Sampling areas of Sparidae specimens (**a**) from Greek seas; Google Map; CM: Chios-Mytilene, CR: Crete, CY: Cyclades, DO: Dodecanese, IO: Ionian Sea, NA: North Aegean Sea, SG: Saronic Gulf, SP: Sporades, TG: Thermaikos Gulf; (**b**) the Mediterranean Sea according to FAO geographical classification of the Mediterranean Sea (area 37). Division captions: 37.1.1 Balearic; 37.1.2 Gulf of Lions; 37.1.3 Sardinia; 37.2.1 Adriatic; 37.2.2 Ionian; 37.3.1 Aegean; (**c**) eastern Atlantic Ocean according to FAO geographical classification; 34.1 Atlantic Ocean Northern Coastal.

**Figure 2 genes-14-01255-f002:**
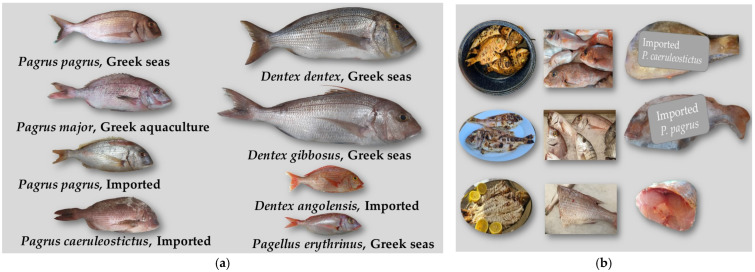
Representative samples for the Sparidae family collected for this study. (**a**) Whole fish collected from Greek fisheries and Greek market. (**b**) Examples of frozen or filleted samples sold in the Greek market and cooked samples served at Greek restaurants.

**Figure 3 genes-14-01255-f003:**
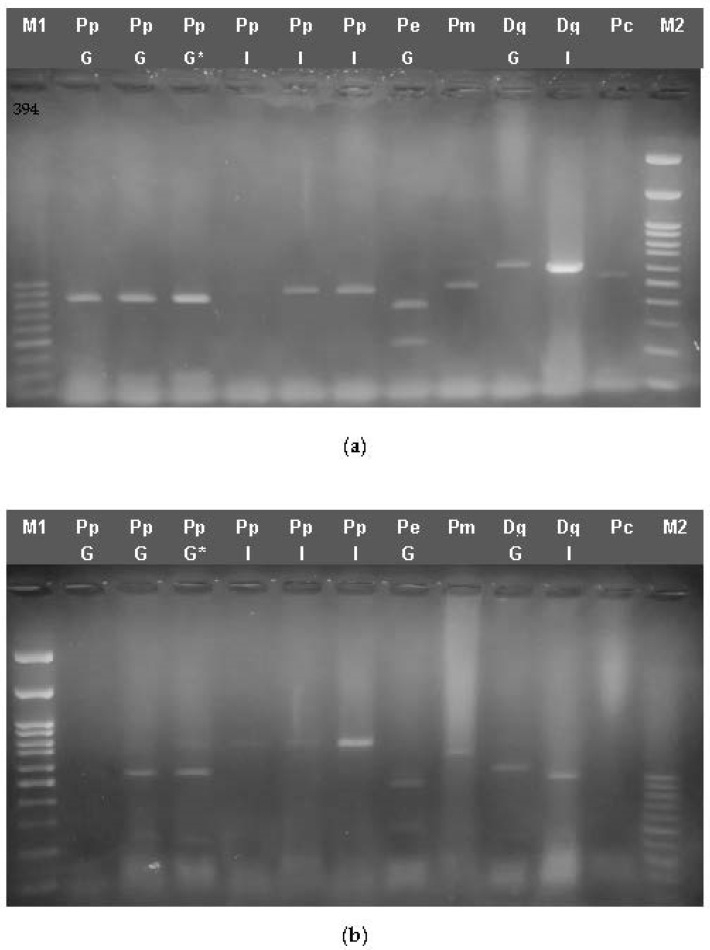
Restriction Enzyme Fragment length polymorphism (RFLP) for mtDNA fragments from Pagrus and other Sparidae species: (**a**) *cytb* (583 bp) digested with Sau3AI (**b**) *COI* barcode (706 bp) digested with Hind III. (**c**) CR (1292 bp) digested with XbaI. Pc: *P. caeruleostictus,* Pp: *P. pagrus*, Pm: *P. major*, Pe: *P. erythrinus*, Dg: *D. gibbosus*, G: fish caught in Greek seas, I: imported in Greek markets. M1: 50 bp ladder, M2: 100 bp ladder, *: cooked samples.

**Figure 4 genes-14-01255-f004:**
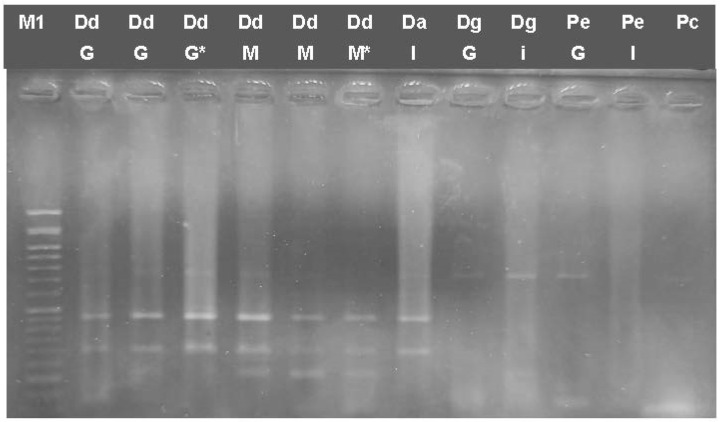
Restriction Enzyme Fragment length polymorphism (RFLP) for mtDNA fragments from *Dentex* sp. and other Sparidae species: COI barcode (706 bp) digested with HindIII. Dd: *D. dentex*, Da: *D. angolensis*, Dg: *D. gibbosus*, Pe: *P. erythrinus*, Pc: *P. caeruleostictus,* G: fish caught in Greek seas, S: Spain. M: 50 bp ladder, G: fish caught in Greek seas, I: imported in Greek markets, M: fish caught in the Mediterranean Sea. M1: 50 bp ladder, *: cooked samples.

**Figure 5 genes-14-01255-f005:**
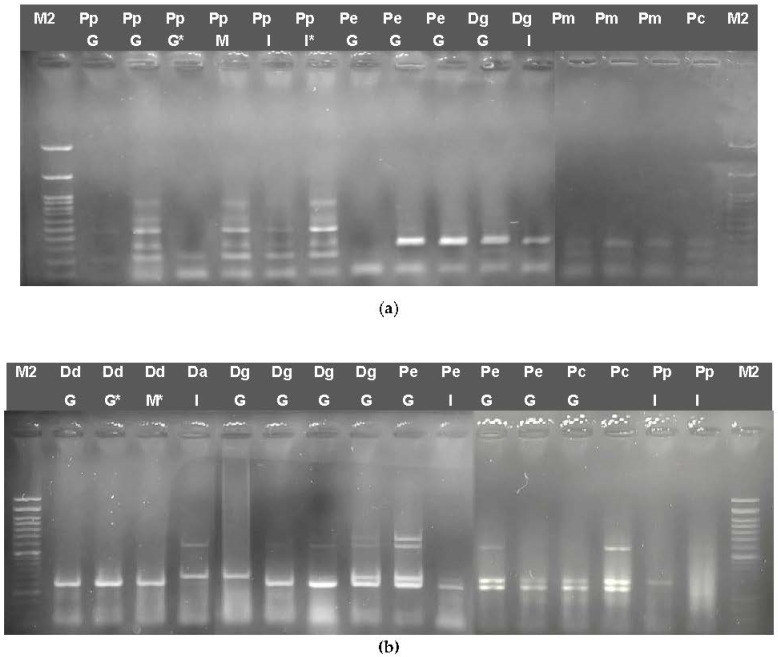
Multiplex PCR reactions electrophoresis for the Sparidae species. (**a**) For the detection of *P. pagrus*; Pp: *P. pagrus*, Pm: *P. major*, Pc: *P. caeruleostictus*, Pe: *P. erythrinus*, Dg: *D. gibbosus*, (**b**). For the detection of *D. dentex*; Dd: *D. dentex*, Da: *D. angolensis*, Dg: *D. gibbosus*, Pe: *P.s erythrinus*, Pc: *P. caeruleostictus.* G: fish caught in Greek seas, I: imported in Greek markets, M: fish caught in the Mediterranean Sea, M2: 100 bp ladder, *: cooked/frozen samples.

**Figure 6 genes-14-01255-f006:**
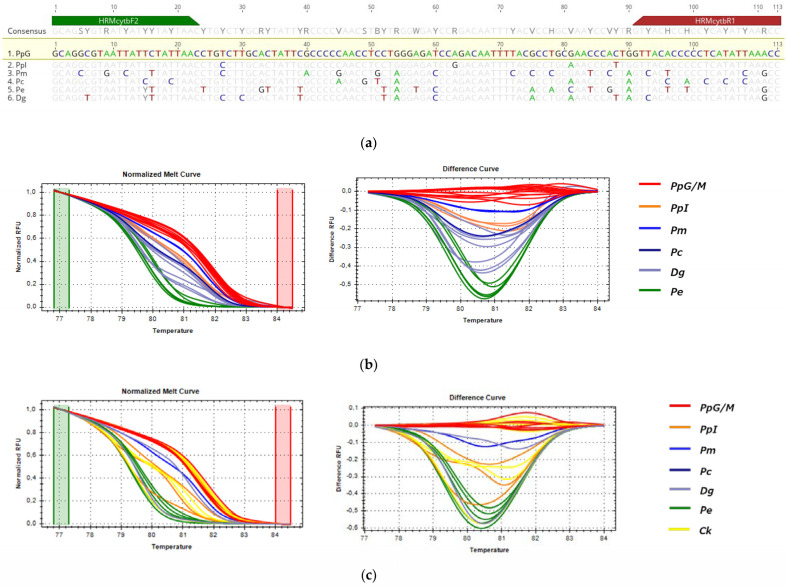
HRM reactions for the discrimination of *P. pagrus* from Greek fisheries (reference cluster in red) compared from other Sparidae species sold in the Greek market with cytb fragment: (**a**) alignment of the tested fragment for all used species; (**b**) HRM analysis of fresh and frozen samples; (**c**) HRM analysis of fresh, frozen and cooked samples; Pp: *P. pagrus*, Pm: *P. major*, Pc: *P. caeruleostictus,* Pe: *P. erythrinus*, Dg: *D. gibbosus*, Ck: cooked samples, G: fish caught in Greek seas, I: imported in Greek markets, M: fish caught in the Mediterranean Sea.

**Figure 7 genes-14-01255-f007:**
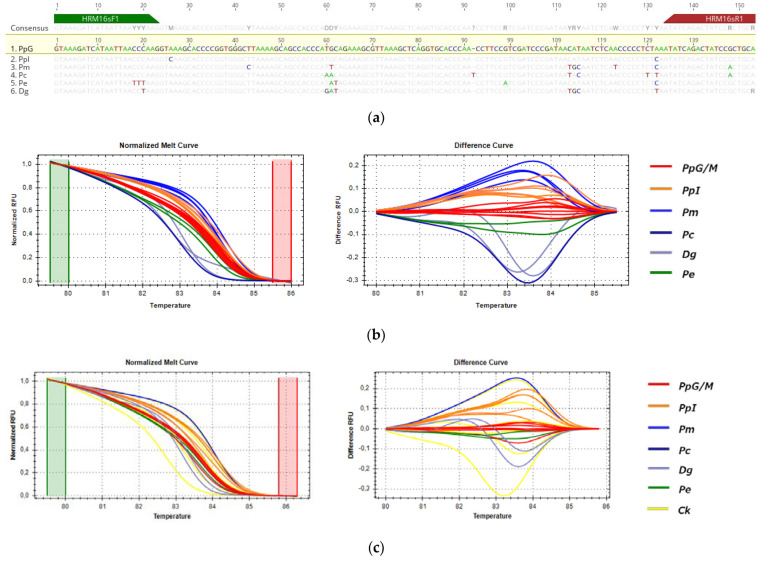
HRM reactions for the discrimination of *P. pagrus* from Greek fisheries compared to other Sparidae species sold in the Greek market with 16s fragment: (**a**) alignment of the tested fragment for all used species; (**b**) HRM analysis of fresh and frozen samples; (**c**) HRM analysis of fresh, frozen and cooked samples; Pp: *P. pagrus*, Pm: *P. major*, Pc: *P. caeruleostictus*, Pe: *P. erythrinus*, Dg: *D. gibbosus*, Ck: cooked samples, G: fish caught in Greek seas, I: imported in Greek markets, M: fish caught in the Mediterranean Sea.

**Figure 8 genes-14-01255-f008:**
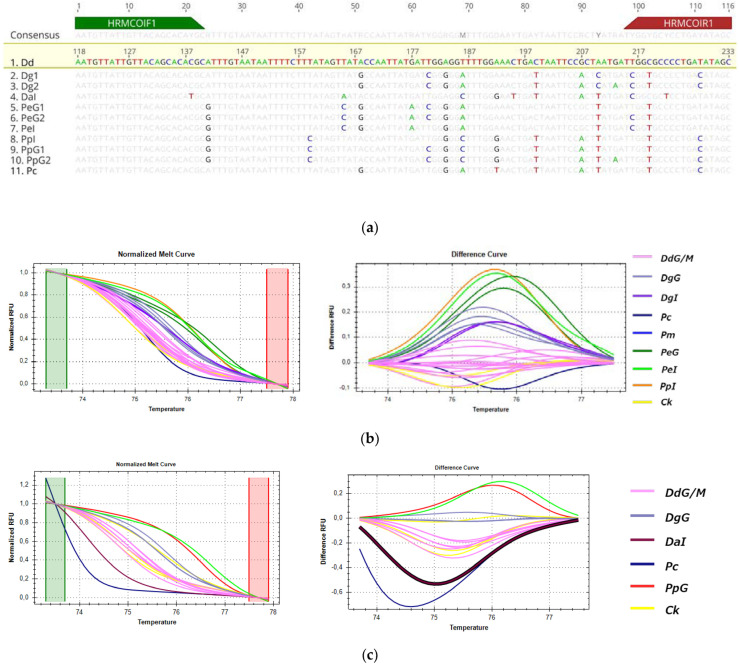
HRM reactions for the discrimination of *D. dentex* from Greek fisheries (reference cluster in pink) compared to other Sparidae species sold in the Greek market using the COI fragment: (**a**) alignment of the tested fragment for all used species; (**b**) HRM analysis of fresh, frozen and cooked samples; (**c**) HRM analysis of fresh and cooked samples; Dd: *D. dentex*, Dg: *D. gibbosus*, Da: *D. angolensis*, Pp: *P. pagrus*, Pm: *P. major*, Pc: *P. caeruleostictus*, Pe: *P. erythrinus*, Ck: cooked samples, G: fish caught in Greek seas, I: imported in Greek markets, M: fish caught in the Mediterranean Sea.

**Figure 9 genes-14-01255-f009:**
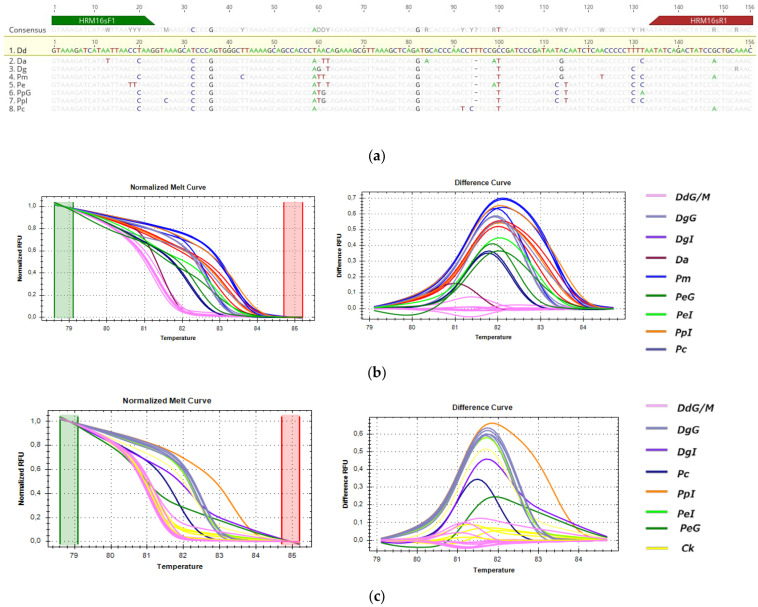
HRM reactions for the discrimination of *D. dentex* from Greek fisheries compared to other Sparidae species sold in the Greek market with 16s fragment: (**a**) alignment of the tested fragment for all used species; (**b**) HRM analysis of fresh, frozen and cooked samples; (**c**) HRM analysis of fresh and cooked samples; Dd: *D. dentex*, Dg: *D. gibbosus*, Da: *D. angolensis*, Pp: *P. pagrus*, Pm: *P. major*, Pc: *P. caeruleostictus,* Pe: *P. erythrinus*, Ck: cooked samples, G: fish caught in Greek seas, I: imported in Greek markets, M: fish caught in the Mediterranean Sea.

**Table 1 genes-14-01255-t001:** List of samples used for molecular identification. CM: Chios-Mytilene, CR: Crete, CY: Cyclades, DO: Dodecanese, IO: Ionian Sea, NA: North Aegean Sea, SG: Saronic Gulf, SP: Sporades, TG: Thermaikos Gulf, ME: Mediterranean Sea, IMP: Imported in the Greek fish market, AQ: Aquaculture products, LFT: lagoon fish traps. Fish were either fished or collected directly from fishermen (FC), local fish markets (FM), supermarkets (SM) or restaurants (RE).

Species	Sample Id	Greek Seas/Other Region	FAO Region	Total Number of Samples	FC Out of Total	FM/SM Out of Total	RE Out of Total	Frozen/ Filleted Out of Total	Cooked/ Out of Total
*P. pagrus*	PpG	CY	37.3.1	12	12				2
(Pp)	PpG	SP	37.3.1	25	24	1		1	3
	PpG	SG	37.3.1	11	10		1		2
	PpG	DO	37.3.1	10	8	2			2
	PpG	CR	37.3.1	3	2	1			2
	PpG	TG	37.3.1	1	1			1	1
	PpG	CM	37.3.1	1	1				1
	PpG	IO	37.2.2	8	8			1	2
	PpI	IMP	34.1	10		10		9	3
	PpM	ME	37.1.1	3	3			3	
*P. major*	Pm	AQ		16		14	2	2	2
(Pm)	Pm	LFT		3		3			2
	Pm	CR	37.1.1	1	1			1	1
	Pm	SP	37.1.1	1	1			1	1
*P. caeruleostictus* (Pc)	Pc	IMP		1		1		1	1
*P. erythrinus* (Pe)	Pe	SP	37.1.1	2	1	1			2
	Pe	DO	37.1.1	8	4	3			3
	PeI	IMP	34.1	1		1		1	1
*D. dentex*	DdG	CY	37.3.1	6	4	1	1		2
(Dd)	DdG	SP	37.3.1	3	2	1			1
	DdG	SG	37.3.1	7	5	1	1		2
	DdG	DO	37.3.1	2	2				2
	DdG	CR	37.3.1	6	6				4
	DdG	ΝA	37.3.1	4	4				2
	DdG	IO	37.2.2	29	29				6
	DdM	ME	37.1.1	3	3			3	
*D. gibbosus*	Dg	SP		2	1	1		1	1
(Dg)	Dg	CY		3	1	2			1
	Dg	IO		2	1	1		2	1
	Dg	SG		2			2	1	1
	DgI	IMP	34.1	3		3		3	2
*D. angolensis* (Da)	Da	IMP	34.1	2		2			1

**Table 2 genes-14-01255-t002:** Primers used for PCR amplifications and sequencing of Sparidae mtDNA.

Name	Primer	Tm °C	
COIpp	TCAACCAACCATAAAGACATCGGCAC	63.2	primers from [37] modified in this study
COIUnR	TAGACTTCTGGGTGGCCRAARAAYCA	64.0	primers from [50]
cytbF1	CATGCTAACGGAGCATCCTTCT	60.3	This study
cytbUR	GCAAATAGGAARTATCAYTCRGG	58.0	This study
16sF2	AGTATGRGCGACAGAAAAGGA	56.9	This study
16SR2	GATTCGGTGGTTGGTCCGTTC	61.8	This study
cytbF2	CATATTAAACCCGAATGATATTT	51.7	This study
CRR	GGGAAGAAACAGCATATTATG	54	This study
COI4FU	CTAGGCGACGACCAGATTTATAATGT	60.8	This study
COIR3	ACTCCTGAGGAGGCAAGTAGG	61.8	This study
COIR4U	GTTAGGTCTACTGATGCTCCTGC	62.4	This study
COIR5	GACTGGCAGGGACAGAAGG	61.0	This study
COIIF	CGCCTAAACCAAACAGCATTC	58.4	This study
ATP6R	GTAAAGGTRTAAGGGAGGAG	55.5	This study
ND2Fd	CACCCTAGCTATCCTCCCCCTCATAGC	66.9	primers from [51]
ND2Rd	AATAACTTCGGGGAGTCACGAGTGTAGG	65.7	primers from [51]
HRMCOIF1	AATGTTATTGTTACAGCACACGC	57.1	This study
HRMCOIR1	GCTATGTCAGGGGCACCAA	58.8	This study
HRMcytbF2	GCAGGCGTAATTATTCTATTAAC	55.3	This study
HRMcytbR1	GGTTTAATATGAGGGGGTGTAAC	58.9	This study
HRM16sF1	GTAAAGATCATAATTAACCCAAG	53.5	This study
HRM16sR1	GTTTGCAGCGGATAGTCTGATAT	58.9	This study

**Table 3 genes-14-01255-t003:** Primers used for PCR amplification and sequencing of Sparidae mtDNA. Positions are indicated.

GENE	Forward Primer	Reverse Primer	Amplicon Length	Annealing T/°C	Restriction Enzyme
*COI*	COIpp	COIUnR	706	54	HindIII or Sau3AI
*cytb*	cytbF1	cytbUR	583	54	Sau3AI
CR	cytbF2	CRR	1292	53	XbaI

**Table 4 genes-14-01255-t004:** Primers used for multiplex PCR amplification of mtDNA fragments of Sparidae.

Species	Forward Primer	Reverse Primer	Length (bp)	Annealing T °C
*P. pagrus* vs. other Sparidae species	COI4FU	COIR5	461	57
	COI4FU	COI4RU	295	
	COI4FU	COIR3	211	
*D. dentex* vs. other Sparidae species	COIIF	ATP6R	673	52
	cytbF1	cytbUR	583	
	COI4FU	COI4RU	295	
	ND2Fd	ND2Rd	249	

**Table 5 genes-14-01255-t005:** Primers used for HRM analysis of Sparidae mtDNA fragments.

GENE	Forward Primer	Reverse Primer	Fragment Length (bp)	Annealing T °C
*COI*	HRMCOIF1	HRMCOIR1	116	53
*cytb*	HRMcytbF2	HRMcytbR1	113	52
*16s*	HRM16sF1	HRM16sR1	156	53

## Data Availability

The data presented in this study are available in the GenBank NIH genetic sequence database under the accession numbers: OQ208746-OQ208756, OQ211305-OQ211329, OQ272113-OQ272123, OQ865594-OQ865603, OQ208828-OQ208830, OQ281602, OQ272124, OQ281603-OQ281609, OQ860775-OQ860777, OQ860773, OQ860774, OQ888163, OQ888164, OQ861107, OQ861152-OQ861162, OQ890738, OQ862795-OQ862820, OQ880490-OQ880493, OQ880557-OQ880559, OQ888790, OQ892284-OQ892292, OQ915022-OQ915035, and OQ903885-OQ903890.

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
