# Peer review of "High-Resolution Melting (HRM) Analysis for Rapid Molecular Identification of *Sparidae* Species in the Greek Fish Market"

_genes, 2023, doi:10.3390/genes14061255_

Round 1

Reviewer 1 Report

The close morphological resemblance of P. pagrus and D. dentex with other local or imported Sparidae species of lower price made it possible that the latter species might be used in substitution events with economical profit. To protect the consumers’ preference for local products, it is meaningful to develop a rapid method for species identification within the Sparidae family. Authors utilized COI barcoding, conventional PCR methods (PCR-RFLP and multiplex PCR) and Real-Time PCR coupled with HRM analysis to identify the P. pagrus and D. dentex from Greek fisheries. Overall, the paper is clearly written and well organized. To help the authors to improve the quality of this paper, some suggestions are given as follows:

 1.       HRM analysis was performed on CO1, Cytb and 16S, however, only two genes’ HRM analysis were provided. Could authors provide all the HRM analysis of three target genes?

2.       Line 120-122, authors mentioned that the mitochondrial DNA evolves much faster than nuclear DNA, my question it that if a single or multiple sequences among the HRM-related region mutated, the Real-Time PCR coupled with HRM analysis will fail to distinguish the target species. Accordingly, it seems better for authors to use genomic DNA in developing similar HRM method.

3.       Line 254, how many clones for a single sample are sequenced? Detailed information should be added.

4.       Line 346, DNA of high quality (260/280 ratio higher than 1.8)…In my opinion, high quality of DNA always refers to 260/280 ratio = 1.8 while 260/230 ratio higher than 2.0. Please check.

5.       The qualities of Figure 3-5 are not suitable for publication, please improve them.

6.       The manuscript should be checked carefully to avoid typing errors.

Line 223, 0.8 should be changed to 0.8%?

Line 627, 400 should be 400 bp?

Line 672, un should be an?

Author Response

  1. HRM analysis was performed on CO1, Cytb and 16S, however, only two genes’ HRM analysis were provided. Could authors provide all the HRM analysis of three target genes?

Response: All three fragments were used in HRM using the same primers. However, we did not present the results that were not informative for each species. As described in Lines 465-467 “all fragments were tested, but two fragments, cytb (113 bp) and 16s (156 bp), were in-formative for the comparison” for P. pagrus and lines 515-516 “two fragments COI (Figure 9) and 16s (Figure 10) were chosen for the analysis” for D. dentex. Text has been corrected to be more precise according to your comment. L646

  1. Line 120-122, authors mentioned that the mitochondrial DNA evolves much faster than nuclear DNA, my question it that if a single or multiple sequences among the HRM-related region mutated, the Real-Time PCR coupled with HRM analysis will fail to distinguish the target species. Accordingly, it seems better for authors to use genomic DNA in developing similar HRM method.

Response: mitochondrial DNA has the ability to discriminate between species for the reasons mentioned in lines 138-141. Although mt genes have different evolutionary rates, intraspecific homology is very high for specific genes.  The method in the current study is optimized for the identification of P.pagrus and D. dentex from the Mediterranean compared to Atlantic populations or other species. To our knowledge, and according to sequences obtained in this study, and comparisons against databases for the region used for HRM analysis, all P. pagrus samples from the Mediterranean show 100% homology while many SNPs differentiate the closely related Sparidae species and 4 SNPs the Mediterranean-Atlantic populations as shown in figure 6a and 7a and D. dentex from other species as shown in figures 8a and 9a.  In the case of the occurrence of a single mutation, HRM might give a false clustering only in the case of a Mediterranean P. pagrus with the imported samples. In any case the results of a suspected sample will be confirmed by sequencing.  Although mtDNA evolves fast there is I very low possibility that multiple mutations will occur simultaneously in the small region examined that with give false positive results for the species under examination.

  1. Line 254, how many clones for a single sample are sequenced? Detailed information should be added.

 Response: thank you, well spotted, we have added the number of samples

  1. Line 346, DNA of high quality (260/280 ratio higher than 1.8)…In my opinion, high quality of DNA always refers to 260/280 ratio = 1.8 while 260/230 ratio higher than 2.0. Please check.

Response: from the literature (Mathiesonand Thomas 2013; Lutz et al. 2023) DNA purity measured by the A260/A280 ratio,has a value around 1.8-2.0 indicating pure DNA and the ideal value for 260:230 ratio is 2.0-2.2. We have both parameters available and checked fro our samples. I will add the second ratio at the M&M too.

  1. The qualities of Figure 3-5 are not suitable for publication, please improve them.

Response: Figures were replaced according to your suggestion

  1. The manuscript should be checked carefully to avoid typing errors.

Response: The manuscript was thoroughly edited for typing errors according to your suggestion

Line 223, 0.8 should be changed to 0.8%?

Response: well spotted typing error was corrected L320

Line 627, 400 should be 400 bp?

Response: well spotted typing error was corrected

Line 672, un should be an?

Response: well spotted typing error was corrected

Reviewer 2 Report

The manuscript “HRM analysis for rapid molecular identification of Sparidae species in the Greek fish market” by Chatzoglou et al., narrates a rapid method for species identification within the Sparidae family. The manuscript covers a very interesting topic. The experiments are well designed and results are nicely depicted.  

A potential increase in fish mislabeling and/or fraud over the last decade is a real concern. In frozen condition or when processed (dried, salted, smoked, canned, etc.), it is very difficult to differentiate similar species based on morphological characteristics. Therefore, it is very crucial to device some easy tools to differentiate closely related species.

The authors mentioned that two members of the Sparidae family, the red porgy Pagrus pagrus and the common dentex, Dentex dentex, are widely traded species in the Greek market with high commercial value. P. pagrus has high commercial potential in Greek market and due to its morphological similarity with some species of the same genus, incidents of mislabeling and fraud have been reported. Same is true for D. dentex also. I agree with the authors that “The higher commercial value of wild P. pagrus and D. dentex, fished in the Greek seas, compared to imported, reared counterparts or other species of the  Sparidae family such as D. gibbosus or the common pandora P. erythrinus, raises the need  for correct and accurate identification of the species and proper labelling of the fish”

Due to limitations of morphological or protein-based fish species identification, DNA-based methods have emerged and proven to give accurate and reliable results. DNA barcoding based on COI gene is the most popular tool for fish species identification and has been adopted by the FDA for the identification of fish products. Only limitation of COI based DNA barcoding is that they are expensive and time consuming. In the present work, the authors have narrated an accurate, rapid, cost-effective method of Sparidae species identification.  The gaps in knowledge as well as objectives of the study are very clear and the manuscript addresses a very important issue. Overall, the manuscript is very well written and very interesting for the readers. I have some suggestions for further improvement of the manuscript.

Introduction: Through it is well structured with lot of information, it looks very lengthy. The authors are requested shorten it keeping the main theme of the manuscript intact.

Materials and Methods: Regarding morphological identification, whether help of any taxonomist was taken or it was done by the authors? If any expert taxonomist was involved, he/she should be acknowledged.

“DNA concentration and quality were assessed using Bio-222 photometer D30 (Eppendorf) and run on 0.8 agarose gel” : should be 0.8%

Results and discussions look good but some minor spell checks are required.

Overall, the manuscript is very relevant, well written and addresses a major concern and has merit to be published in a reputed journal like “Genes”

Some minor spell check required especially species name should be italics. 

Author Response

Introduction: Through it is well structured with lot of information, it looks very lengthy. The authors are requested shorten it keeping the main theme of the manuscript intact.

Response: the introduction has been modified according to your suggestion.

Materials and Methods: Regarding morphological identification, whether help of any taxonomist was taken or it was done by the authors? If any expert taxonomist was involved, he/she shouldbe acknowledged.

Response: the morphological identification has been performed by the co- authors: N. Tsaoussi, G. Triantaphyllides and H. Miliou who are experienced  ichthyologists and fish biologists.

“DNA concentration and quality were assessed using Bio-222 photometer D30 (Eppendorf) andrun on 0.8 agarose gel” : should be 0.8%

Response: well spotted, typing error was corrected. L320

Results and discussions look good but some minor spell checks are required.

Response: The manuscript was thoroughly edited for typing errors according to your suggestion

Reviewer 3 Report

Genes-2414091

Chatzoglou et al

In this paper, several methods to identify Sparidae species in the Greek fish market are presented. The goal of this study (as stated by the authors, lines 29-30) is “to develop a rapid method for species identification within the Sparidae family.” Of the methods presented, HRM analysis shows high precision and repeatability, and revealed some labelling errors in the analysed samples.

General comments

The justification of the work is adequate. The MS is well written and interesting, but I have some concerns that need to be addressed before recommending its publication.

The paper is perhaps too long and contains a lot of information, some of which could be summarized.

For readers unfamiliar with the Greek market, it is complicated to follow the origin of the samples and relate them to the results. It would be appreciated if the authors could find a way to make it more traceable (matmet-results-discussion). I apologize in advance to the authors if it is me, that I am unable to relate the analyses to the origins of the samples.

Introduction

The introduction is very comprehensive. References cited are to reviews or specific to the family Sparidae. In my opinion, it could be completed with some quotations of similar situations in another family or genus of fish that have been used to detect mislabelling percentages.

Line 117. Please rephrase it (“thus sequences thus avoiding…”

Line 128. Please change “… un unknow fish…” by “… an unknow fish…”

Line 129. Please add the number (27) for de reference [Ward]

Material and Methods

In the first paragraph of MatMet, the authors describe the origin of the 106 Pagrus spp. and 57 D. dentex individuals and refer to table 1 for further information. Despite the information in the above table and maps (Figure 1) it is not clear how many individuals of each type are analysed. For example, in the first row of table 1, the total number of samples is 12 and the source FC/FM/RE, if it is indicated how many correspond to each origin it would be possible to calculate percentages of mislabelling in the total.

As the authors state in line 157 that it is possible to distinguish aquaculture specimens from fishery specimens, for readers unfamiliar with the Greek fish market it would be interesting to show this information in a more clarifying way.

Table 1: Please, list the sampling areas in alphabetical order or from west to east or at least in the same order as they are listed in the table. It is difficult to identify them. Also, "a, b and c" do not appear in the table but do appear in the title.

It would be interesting to highlight in some way which of the samples come from aquaculture.

Lines 231- 232. Authors said that “annealing temperatures were adjusted according to the Tm of primers (Table 2)”. However, what is observed is that all PCRs were carried out at a temperature of 54°C regardless of the Tm of the primers. Please change the text accordingly.

Table 2: Please remove "positions are indicated" from the title as there are no positions in the table. The same goes for Table 3. The pair of primers cytbF1 and cytbUR is repeated. Please remove it. In the last column when changing primers to primer, as the quote refers to a single primer.

There are two Tables 3. These tables, including table 4 are confusing. I believe that table 2 and the second table 3 should be merged into a single table, which includes all the primers used. There are ways to differentiate the two types of primers, if that is of concern to the authors.

The same with tables 3 (first) and 4, they could be joined together and it would be easier to follow.

I am concerned about the high concentration of MgCl2 used in all amplifications. At these concentrations, unspecificity is high. Have the authors tested lower concentrations? As the annealing temperature is low and magnesium is high, it would be interesting to do more amplification tests to confirm repeatability.

Results

Line 320: Supermarket

Line 361: P. pagrus in italics

RFLP analysis. As indicated in line 284 RFLP products were visualized on 1.2% agarose gels. In this type of approach, it is recommended to use agarose mixtures that increase the resolution and a higher concentration (e.g. 3%) as in Pérez et al, 2005. J. Agric. Food Chem. 2005, 53, 13 for example.

The alignments in figures 6, 7, 9 and 10 should be supplementary tables or presented in a better format that allows easy comparison of the information provided.

Figures 8 and 11 should be tables, for better visualisation of the information, or add some value in the text and the rest as complementary information.

Discussion

As I mentioned in the introduction section, I think the discussion would benefit for comparing similar cases in other species groups (if possible).

Moreover, I am missing more concrete data on “mislabelling incidents” See for instance lines 590, 592

Lines 627-633 refer to a single reference, the authors could enrich this paragraph with more references.

Lines 655-656 refer to aquaculture and lagoon fish traps. Again, it is not very clear to me which of the samples they are referring to.

Author Response

General comments

The justification of the work is adequate. The MS is well written and interesting, but I have some concerns that need to be addressed before recommending its publication.

The paper is perhaps too long and contains a lot of information, some of which could be summarized.

Response: the paper has been summarized (mainly introduction), according to your suggestion.

For readers unfamiliar with the Greek market, it is complicated to follow the origin of the samples and relate them to the results. It would be appreciated if the authors could find a way to make it more traceable (matmet-results-discussion). I apologize in advance to the authors if it is me, that am unable to relate the analyses to the origins of the samples.

Response: Table 1 has been reorganized to be more comprehensive. Please let us know if further clarifications are needed

Introduction

The introduction is very comprehensive. References cited are to reviews or specific to the family Sparidae. In my opinion, it could be completed with some quotations of similar situations in another family or genus of fish that have been used to detect mislabelling percentages.

Response: according to your suggestion, some references on the specific topic has been added, however they were placed in the discussion section as the introduction was too lengthy.

Line 117. Please rephrase it (“thus sequences thus avoiding…”

Response: well spotted, syntax error was corrected

Line 128. Please change “… un unknow fish…” by “… an unknow fish…”

Response: well spotted, typing error was corrected. L148

Line 129. Please add the number (27) for de reference [Ward]

Response: well spotted, typing error was corrected. L149

Material and Methods

In the first paragraph of MatMet, the authors describe the origin of the 106 Pagrus spp. and 57 D. dentex individuals and refer to table 1 for further information. Despite the information in the above table and maps (Figure 1) it is not clear how many individuals of each type are analysed. For example, in the first row of table 1, the total number of samples is 12 and the source FC/FM/RE, if it is indicated how many correspond to each origin it would be possible to calculate percentages of mislabelling in the total.

Response: Table was formatted according to your comments in an effort to be more comprehensive. Please let us know if further clarifications are needed

As the authors state in line 157 that it is possible to distinguish aquaculture specimens from fishery specimens, for readers unfamiliar with the Greek fish market it would be interesting to show this information in a more clarifying way.

Response: The sentence was rephrased in a more clarifying way. More information is given in the discussion

Table 1: Please, list the sampling areas in alphabetical order or from west to east or at least in the same order as they are listed in the table. It is difficult to identify them. Also, "a, b and c" do not appear in the table but do appear in the title.

It would be interesting to highlight in some way which of the samples come from aquaculture.

Response: Table was formatted and completed according to your comments in the same order as they appear in Table 2

Lines 231- 232. Authors said that “annealing temperatures were adjusted according to the Tm of primers (Table 2)”. However, what is observed is that all PCRs were carried out at a temperature of 54°C regardless of the Tm of the primers. Please change the text accordingly.

Response: well spotted, the text was changed to be more comprehensive

Table 2: Please remove "positions are indicated" from the title as there are no positions in the table. The same goes for Table 3. The pair of primers cytbF1 and cytbUR is repeated. Please remove it. In the last column when changing primers to primer, as the quote refers to a single primer.

Response: well spotted, the text was removed

There are two Tables 3. These tables, including table 4 are confusing. I believe that table 2 and the second table 3 should be merged into a single table, which includes all the primers used. There are ways to differentiate the two types of primers, if that is of concern to the authors.

The same with tables 3 (first) and 4, they could be joined together and it would be easier to follow.

Response: The second Table 3 was mistyped and is “Table 5” describing HRM PCR amplification conditions. Table 2 includes all primers used in PCR reactions, presenting their sequence Tm and reference. Primers sequences from Table 5 were transferred to Table 2 in order to have all information in one table. However, we believe that we should keep Tables 3, 4 and 5 separately, describing PCR conditions and fragment lengths used in each method in order to be more informative.

I am concerned about the high concentration of MgCl2 used in all amplifications. At these concentrations, unspecificity is high. Have the authors tested lower concentrations? As the annealing temperature is low and magnesium is high, it would be interesting to do more amplification tests to confirm repeatability.

Response: From the literature, and official methods, the proposed concentration for COI barcoding is 2,5 mM. (Handy et al. 2011, Landi et al. 2014) and temperature 55°C. Initial PCR reactions were performed under these conditions. However, the amplification was not always efficient, due to primer mismatch for the samples of the Sparidae family. Therefore, different concentrations of MgCl2 were tested (2-3 mM) combined with various annealing temperatures (52-55°C). The final PCR conditions   presented in M&Ms were adjusted to ensure amplification of the PCR products without the presence of unspecific products. Moreover, KAPA HRM protocol suggests to use 2,5 mM MgCl2.

Line 320: Supermarket

Response: it refers to various supermarkets in Greece

Line 361: P. pagrus in italics

Response: well spotted, typing error was corrected

RFLP analysis. As indicated in line 284 RFLP products were visualized on 1.2% agarose gels. In this type of approach, it is recommended to use agarose mixtures that increase the resolution and a higher concentration (e.g. 3%) as in Pérez et al, 2005. J. Agric. Food Chem. 2005, 53, 13 for example.

Response: In RFLP analysis used in this study, fragments of various lengths were used so we decided to use 1.2% agarose gel to estimate the band pattern. In the literature in RFLP analysis agarose gels of various concentrations are used ranging from 1-4% (Ferreira et al. 2014, Wolf et al. 1999, Cocolin et al. 2000).  The markers used (50 and 100 bp ladders) had clear separation in 1.2% agarose gel. All bands were identified, therefore, we didn’t switch to more concentrated gel.

The alignments in figures 6, 7, 9 and 10 should be supplementary tables or presented in a better format that allows easy comparison of the information provided.

Response: The alignments figures were enlarged but we kept the figures in the figures, for easier visualization by the reader in relation with the text. However if you believe that the results are still not visible enough, we can provide them as supplementary materials.

Figures 8 and 11 should be tables, for better visualisation of the information, or add some value in the text and the rest as complementary information.

Response: The figures were moved to the supplement

Discussion

As I mentioned in the introduction section, I think the discussion would benefit for comparing similar cases in other species groups (if possible).

Response: More references are added to the discussion

Moreover, I am missing more concrete data on “mislabelling incidents”. See for instance lines 590, 592

Response: well spotted, number of samples are provided in this section.

Lines 627-633 refer to a single reference, the authors could enrich this paragraph with more references.

Response: More references are added in this paragraph.

Lines 655-656 refer to aquaculture and lagoon fish traps. Again, it is not very clear to me which of the samples they are referring to.

Response: tables and samples id have been re-assigned in order to be more comprehensive. Please let us know if it remains unclear, the samples’ references.

Reviewer 4 Report

In the manuscript entitled “HRM analysis for rapid molecular identification of Sparidae species in the Greek fish market” authors described interesting new methods for tracking fish frauds in Greek market. The manuscript shows interesting results, however due to methodological concerns I do not recommend the manuscript to be published in Genes. The detailed justification of final decision is provided in the comments below.

Please note L refers to line.

Major concerns

-     Overall, article is interesting and provides new methodologies for tracking Sparidae frauds, however I have doubts about methodologies. First, based on provided by authors Figures for Multiplex PCR, one can get impression, that reaction was not fully optimized. For instance, in L428 authors mentioned “three-band pattern of 461, 295 and 211 bp”, while electrophoresis showed 6 or 7 band patterns (which is also difficult to justify due to Figure quality), which indicate non-specific amplification. Therefore, such methodology cannot be accepted for differencing species. 

-     For HRMA authors did not perform cross-laboratory testing, which questions the repeatability of the designed test. Authors claimed that they ensured repeatability by using different sample types (fresh, frozen, cooked) (L615-616) but this is not how repeatability is tested, check for example Monteiro et al. 2021 doi: 10.1016/j.fct.2021.112440. The repeatability of the research is crucial, when presenting new methodologies, especially when considering methods that have potential to be implemented in rapid fraud tracking. Moreover, I did not find information on technical repetitions of tested samples. According to MIQE guidelines, qPCR should be run in such replicates at least 2-3, ideally 5. Please provide such information in the text. Additionally, please provide amplification curves and amplification efficiency for each species (e.g., in supplementary file). Was the Melting curve shape sensitivity and Tm difference threshold set by the software automatically or authors set it manually (L477-478; L489-490; L518-519; L532-533)? This is crucial information to replicate the experiment.

-     RFLP analysis seems well designed and provides the accurate results, however, full judgement is unable due quality of the Figures. Please provide better quality gel photos, e.g., Figure 4 seems to be trimmed too much in the bottom.

-     L231 This is misleading. Table 2 provides two temperatures (one for each primer), therefore in this form experiment is impossible to repeat by other researchers. Please elaborate.

-     The manuscript is too wordy in some parts (Introduction, Materials and Methods, Results) and superficial in Discussion part, e.g.: 

·      L149-152 basic information about HRM that can be omitted,

·      Manuscript readability can benefit from giving the specific samples codes (like in Figures description) instead of constantly repeating “from Greek market”, “from Mediterranean Sea” etc.

·      L340-349 This can be described in M&M with 1-2 sentences,

·      L389-391 This are Materials and Methods not Results, can be omitted,

·      L426-427 M&M not Results, please exclude,

·      L444-462 M&M not Results, please change,

·      L596-606 Off topic not relevant for the manuscript, please change,

·      L638-643 Please focus on discussing your results, this is M&M description of HRM for third time.

·      L674-682 Please elaborate on that matter.

Minor concerns

L36 Keywords should not duplicate the words from title to ensure better visibility. Please change the keywords that duplicate the title i.e., HRM, Sparidae.

L114 According to MIQE guidelines RT stands for Reverse Transcription. Please change to qPCR.

L128 Please change “un” to “an”.

L129 Please provide correct reference in brackets “[ward]” or remove.

L164 spp. does not appear in italic. Please change.

L215 (Table 1) Please change the headings to be more informative e.g. number of frozen/ filleted samples out of total; number of cooked samples out of total.

L223 Please add % after “0.8”.

L265 According to Guidelines for Formatting Gene and Protein Names, fish genes should be named lowercase and italic. Please change throughout the manuscript.

L361 P. pagrus should be in italic. Please change.

L379 “…, thus considering it as unintentional mislabeling” – this is discussion not description of results. Please remove.

L395 is this the correct size? 530+150=680 bp, for other species it was 580 bp. Please check.

L478 Manuscript could benefit from showing confidence levels for each sample. Please consider it.

L505 (Figure 8) Please consider moving Figure to supplement.

L508-511 This seems more like discussion, please consider moving to Discussion part.

L551 (Figure 11) Please consider moving Figure to supplement.

L579 “that” should not be in italic.

L627 Please add “bp” after “(>400”

L627-628 Were there any signs of DNA degradation? In L223 authors claimed they assessed quality of DNA on 0.8% gel, but in the text, they did not mentioned degradation (L346 “…, DNA of high quality … was used in downstream applications…”. Please explain.

Author Response

Major concerns

-

Overall, article is interesting and provides new methodologies for tracking Sparidae frauds, however I have doubts about methodologies. First, based on provided by authors Figures for Multiplex PCR, one can get impression, that reaction was not fully optimized. For instance, inL428 authors mentioned “three-band pattern of 461, 295 and 211 bp”, while electrophoresis showed 6 or 7 band patterns (which is also difficult to justify due to Figure quality), which indicate non-specific amplification. Therefore, such methodology cannot be accepted for differencing species.

Response: Well spotted, the sentence is not clear. Several trials were performed with stricter PCR conditions i.e higher Tm and lower MgCl2. However, when byproducts for P. pagrus disappeared under these conditions, the amplification was not efficient for other species. Since our results showed that HRM was more efficient for achieving the identification of PpG, the reaction was not further optimized as it was time consuming. The sentence is rephrased in results and discussion to be more precise.L628

-

For HRMA authors did not perform cross-laboratory testing, which questions the repeatability of the designed test. Authors claimed that they ensured repeatability by using different sample types (fresh, frozen, cooked) (L615-616) but this is not how repeatability is tested, check for example Monteiro et al. 2021 doi: 10.1016/j.fct.2021.112440. The repeatability of the research is crucial, when presenting new methodologies, especially when considering methods that have potential to be implemented in rapid fraud tracking.

Response: The term “repeatability” to described that all samples of known sequence were identified as expected in all trials used. However, the term “consistency” might be more appropriate. The goal of this study was to compare sequencing-based methods to non-sequencing rapid methods for species identification. HRM was optimized, based on already sequenced samples restricted to Sparidae samples found in the Greek market that could potentially be used on fraudulent actions. It has not yet been applied to extensive market controls. Since HRM seems to be a promising tool against seafood fraud for many species, as a next step we will further validate the method with a series of experiments to express repeatability and reproducibility as well as test its efficacy against more Sparidae or other species.

Moreover, I did not find information on technical repetitions of tested samples. According to MIQE guidelines, qPCR should be run in such replicates at least 2-3, ideally 5. Please provide such information in the text.

Response: Well spotted, samples were run as triplicates in 3 or 4 independent trial each. The information is added in M&M section. L455

Additionally, please provide amplification curves and amplification efficiency for each species (e.g., in supplementary file).

Response: Robust, efficient and specific PCR is critical when results depend on the PCR product melting profile. We used a gradient thermal cycler with different melting temperatures, various MgCl2 concentrations and gel electrophoresis in order to optimize the method for amplification. Furthermore, we photometrically measured the initial concentration of DNA template and normalized to equal quantities per experiment. Strict PCR efficiency estimation is calculated through serial dilutions of template in order to measure differential gene expression by RT-qPCR (mainly with the Pfaffl method), which was not the aim of our study. However, through normalization we achieved the fact that all samples are on the same scale (equal starting points) in order to be comparable. Amplification efficiency was based on the automatic calculation of CFX manager 3.1 QC, to be between 90% and 110%. In each trial, wells that failed this rule were excluded from HRM analysis. Amplification curves and Cq for each species are provided in Figure S1 and Table S1 (Supplementary material).

Was the Melting curve shape sensitivity and Tm difference threshold set by the software automatically or authors set it manually (L477-478; L489-490; L518-519; L532-533)? This is crucial information to replicate the experiment.

Response: the analysis was performed for Melting curve shape sensitivity and Tm difference threshold set by the software automatically (data not shown) however intraspecies variation was not always detectable. As the aim of this study was to distinguish PpG from PpI samples the parameters were set manually according to species identification by sequencing. The values used are provided in L….. of the results. The amplification curves and amplification efficiency for each species (e.g., in supplementary file), are provided in Supplementary materials.

-      RFLP analysis seems well designed and provides the accurate results, however, full judgement is unable due quality of the Figures. Please provide better quality gel photos, e.g., Figure 4 seems to be trimmed too much in the bottom.

Response: well spotted, RFLP photos are replaced according to the Reviewers’ suggestion

-      L231 This is misleading. Table 2 provides two temperatures (one for each primer), therefore in this form experiment is impossible to repeat by other researchers. Please elaborate.

Response: Table 2 provides the Tm temperature of all primers. Tm of each amplification used in different methodologies are described in Tables 3, 4 and 5

-      The manuscript is too wordy in some parts (Introduction, Materials and Methods, Results) and superficial in Discussion part, e.g.:

  • L149-152 basic information about HRM that can be omitted,

Response: Lines have been removed as suggested

  • Manuscript readability can benefit from giving the specific samples codes (like in Figures description) instead of constantly repeating “from Greek market”, “from Mediterranean Sea” etc.

Response: we have made some changes in the manuscript (Tables, descriptions) in order to be more comprehensive, regarding the origin of the samples, according to reviewers’ suggestions.

  • L340-349 This can be described in M&M with 1-2 sentences,

Response: the sentence was moved to M&M as suggested

  • L389-391 This are Materials and Methods not Results, can be omitted,

Response: the sentence has been omitted as suggested

  • L426-427 M&M not Results, please exclude,

Response: the lines have been rephrased according to reviewers’ suggestions

  • L444-462 M&M not Results, please change,

Response: the sentence was moved to M&M as suggested

  • L596-606 Off topic not relevant for the manuscript, please change,

Response: we believe that this sentence is relevant, as one of the main issues in Greek market fraud is the mislabelling of P.major as P. pagrus from wild catch. With the presence of P. major in the Greek seas we have to precise the terms of misalbelling in such cases. The paragraph has been rephrased to be more clear.

L638-643 Please focus on discussing your results, this is M&M description of HRM for third time.

  • Response: well spotted, the sentence was rephrased and summarized

L674-682 Please elaborate on that matter.

Response: the paragraph was enriched with more reference and comments on the matter

Minor concerns

L36 Keywords should not duplicate the words from title to ensure better visibility. Please changethe keywords that duplicate the title i.e., HRM, Sparidae.

Response: “HRM, Sparidae” keywords have been removed and “fraud” has been added. L36

L114 According to MIQE guidelines RT stands for Reverse Transcription. Please change to qPCR.

Response: Error was corrected

L128 Please change “un” to “an”.

Response: Typing error was corrected

L129 Please provide correct reference in brackets “[ward]” or remove.

Response: Typing error was corrected

L164 spp. does not appear in italic. Please change.

Response: The sentence was rephrased according to reviewer’s 3 suggestion

L215 (Table 1) Please change the headings to be more informative e.g. number of frozen/ filleted samples out of total; number of cooked samples out of total.

Response: The table was reformatted according to reviewer’s 3 and your suggestions

L223 Please add % after “0.8”.L320

Response: well spotted, typing error was corrected

L265 According to Guidelines for Formatting Gene and Protein Names, fish genes should benamed lowercase and italic. Please change throughout the manuscript.

Response: Well spotter, the names were changed throughout the manuscript according to your suggestion.

L361 P. pagrus should be in italic. Please change.

Response: Error was corrected

L379 “…, thus considering it as unintentional mislabeling” – this is discussion not description of results. Please remove.

Response: The sentence was moved to the Discussion

L395 is this the correct size? 530+150=680 bp, for other species it was 580 bp. Please check. L575

Response: well spotted, error was corrected and 530 was replaced by 430

L478 Manuscript could benefit from showing confidence levels for each sample. Please consider it.

Response: A figure of confidence levels was added to the Supplementary materials

L505 (Figure 8) Please consider moving Figure to supplement.

Response: The figure was moved to the supplement

L508-511 This seems more like discussion, please consider moving to Discussion part.

Response: The paragraph was removed and its content has been integrated in the discussion part.

L551 (Figure 11) Please consider moving Figure to supplement.

Response: The figure was moved to the supplement

L579 “that” should not be in italic.

Response: Typing error was corrected

L627 Please add “bp” after “(>400”

Response: Typing error was corrected

L627-628 Were there any signs of DNA degradation? In L223 authors claimed they assessedquality of DNA on 0.8% gel, but in the text, they did not mentioned degradation (L346 “…, DNA of high quality … was used in downstream applications…”. Please explain.

Response: the sentence was rephrased in order to be more precise. L978

Round 2

Reviewer 4 Report

I appreciate authors time and effort to answer all my questions and comments. Some minor editorial, issues were still detected:

Table 3 – missing bottom borders.

Table 5 – missing bottom borders.

Fig 3 C – desynchronized labels with electrophoresis lines. 

Line 578 highlighted comma.

Author Response

Thank you for your time for reviewing our manuscript

Regarding the minor editorial issues:

Point 1: Table 3 – missing bottom borders.

Response 1: Table 3 has been corrected

Point 2: Table 5 – missing bottom borders.

Response 2: Borders have been corrected

Point3: Fig 3 C – desynchronized labels with electrophoresis lines. 

Response 3: Fig 3

Point 4: Line 578 highlighted comma.

Response 4: the highlightening has been removed

Moreover, we also noted some minor editorial issues

 L5 author’s name has been corrected

Bottom borders were also corrected for tables 1,2 and 4